# Treatment Patterns, Testing Practices, and Outcomes in Patients with *EGFR* Mutation-Positive Advanced Non-Small-Cell Lung Cancer in Poland: A Descriptive Analysis of National, Multicenter, Real-World Data from the REFLECT Study

**DOI:** 10.3390/cancers15051581

**Published:** 2023-03-03

**Authors:** Adam Pluzanski, Maciej Bryl, Izabela Chmielewska, Grzegorz Czyzewicz, Joanna Luboch-Kowal, Anna Wrona, Agnieszka Samborska, Maciej Krzakowski

**Affiliations:** 1Lung Cancer and Chest Tumors Department, Maria Sklodowska-Curie National Research Institute of Oncology, 02-781 Warsaw, Poland; 2Department of Clinical Oncology with the Subdepartment of Diurnal Chemotherapy, E. J. Zeyland Wielkopolska Center of Pulmonology and Thoracic Surgery, 60-569 Poznan, Poland; 3Department of Pneumonology, Oncology and Allergology, Medical University of Lublin, 20-090 Lublin, Poland; 4Department of Oncology, The John Paul II Specialist Hospital, 31-202 Cracow, Poland; 5Department of Oncology, Lower Silesian Oncology Center, Home Hospice, Wroclaw Medical University, 53-413 Wroclaw, Poland; 6Department of Oncology and Radiotherapy, Medical University of Gdansk, 80-210 Gdansk, Poland; 7AstraZeneca Pharma Poland Sp.zo.o., 02-676 Warsaw, Poland

**Keywords:** non-small-cell lung cancer, EGFR mutation, tyrosine kinase inhibitors, brain metastases, real-world data

## Abstract

**Simple Summary:**

Lung cancer is one of the most frequently diagnosed malignant diseases in Poland. Presented real-world data might help in making a clinical decision on cancer treatment and provide an argument for the fast availability of breakthrough therapy. The REFLECT study is one of the largest real-world studies dedicated to patients with non-small-cell lung carcinoma (NSCLC). The results of presented descriptive analysis of the Polish population of the REFLECT study highlight the need for effective treatments and diagnostics of patients with advanced NSCLC with EGFR mutations.

**Abstract:**

Non-small-cell lung cancer (NSCLC) represents 85% of new cases of lung cancer. Over the past two decades, treatment of patients with NSCLC has evolved from the empiric use of chemotherapy to more advanced targeted therapy dedicated to patients with an epidermal growth factor receptor (*EGFR*) mutation. The multinational REFLECT study analyzed treatment patterns, outcomes, and testing practices among patients with *EGFR*-mutated advanced NSCLC receiving first-line EGFR tyrosine kinase inhibitor (TKI) therapy across Europe and Israel. The aim of this study is to describe the Polish population of patients from the REFLECT study, focusing on treatment patterns and T790M mutation testing practice. A descriptive, retrospective, non-interventional, medical record-based analysis was performed on the Polish population of patients with locally advanced or metastatic NSCLC with *EGFR* mutations from the REFLECT study (NCT04031898). A medical chart review with data collection was conducted from May to December 2019.The study involved 110 patients. Afatinib was used as the first-line EGFR-TKI therapy in 45 (40.9%) patients, erlotinib in 41 (37.3%), and gefitinib in 24 (21.8%) patients. The first-line EGFR-TKI therapy was discontinued in 90 (81.8%) patients. The median progression-free survival (PFS) on first-line EGFR-TKI therapy was 12.9 months (95% CI 10.3–15.4). A total of 54 patients started second-line therapy, of whom osimertinib was administered to 31 (57.4%). Among 85 patients progressing on first-line EGFR-TKI therapy, 58 (68.2%) were tested for the T790M mutation. Positive results for the T790M mutation were obtained from 31 (53.4%) tested patients, all of whom received osimertinib in the next lines of therapy. The median overall survival (OS) from the start of first-line EGFR-TKI therapy was 26.2 months (95% CI 18.0–29.7). Among patients with brain metastases, the median OS from the first diagnosis of brain metastases was 15.5 months (95% CI 9.9–18.0). The results of the Polish population from the REFLECT study highlight the need for effective treatment of patients with advanced *EGFR*-mutated NSCLC. Nearly one-third of patients with disease progression after first-line EGFR-TKI therapy were not tested for the T790M mutation and did not have the opportunity to receive effective treatment. The presence of brain metastases was a negative prognostic factor.

## 1. Introduction

Lung cancer is one of the most frequently diagnosed malignant diseases and the most common cause of cancer death worldwide. In 2020, there were more than 2.2 million estimated new cases of lung cancer globally [1]. In 2019, there were more than 22,000 new cases of lung cancer in Poland. The 5-year lung cancer mortality rates in Poland are among the highest in the European Union [2]. Non-small-cell lung cancer (NSCLC) represents 85% of new cases of lung cancer [3]. Over the past two decades, treatment of patients with NSCLC has evolved from the empiric use of chemotherapy to more advanced targeted therapy dedicated to patients with driver mutations. Epidermal growth factor receptor (*EGFR*) mutations occur in 10–15% of NSCLC cases in Western countries [4]. During REFLECT study enrollment for patients with locally advanced and/or metastatic NSCLC harboring an *EGFR* mutation, the first-line treatment includes first-generation EGFR-tyrosine kinase inhibitors (TKIs) such as erlotinib and gefitinib, second-generation EGFR-TKIs such as afatinib and dacomitinib, and the newest third-generation EGFR-TKI osimertinib. Currently osimertinib is becoming the standard of care [5]. Regrettably, around 60–70% of patients treated with first- or second-generation EGFR-TKIs develop resistance due to the T790M mutation in exon 20 [6,7,8]. Osimertinib is effective in patients who acquired the T790M mutation following first- or second-generation EGFR-TKI therapy [9]. Patients treated with osimertinib as the first-line treatment had a significantly longer median progression-free survival (PFS) and overall survival (OS) than patients treated with first-generation EGFR-TKIs. Osimertinib also has an advantage over first- and second-generation TKIs in the treatment and prevention of central nervous system (CNS) progression [10,11,12]. However, in routine clinical practice, almost 30% of patients who progressed on first- or second-generation EGFR-TKIs do not receive subsequent treatment [13,14]. In Poland, osimertinib has been available as first-line treatment since January 2021, whereas the European Medicines Agency (EMA) approved first-line osimertinib in June 2018 [15]. Outcomes of Polish patients with lung cancer may be improved by earlier adoption of more effective treatments.

Collecting real-world data might help in making a clinical decision on NSCLC treatment and provide an argument for the faster availability of breakthrough therapy. The REFLECT study is one of the largest real-world studies dedicated to patients with NSCLC. The aim of the presented study is to describe the Polish population of patients with *EGFR*-mutated advanced NSCLC receiving first-line EGFR-TKI therapy in the REFLECT study, focusing on treatment patterns and T790M testing practice.

## 2. Materials and Methods

### 2.1. Study Design

REFLECT is a multinational, multicenter, retrospective, non-interventional, medical record-based analysis of patients with locally advanced or metastatic NSCLC with *EGFR* mutations in Europe and Israel (NCT04031898). A medical chart review with data collection was conducted from May to December 2019. 

The primary endpoints of the study are to describe first-line EGFR-TKI therapy, the proportion of patients with disease progression after first-line EGFR-TKI treatment, and estimation of real-world PFS (rwPFS), defined as radiological progression, start of a new line of therapy, death, or clinical progression as evaluated by the physician. Other primary endpoints were treatment strategies in subsequent lines of therapy and the attrition rate of patients who progressed on first-line EGFR-TKI therapy. Secondary endpoints included *EGFR* mutation analysis, with a focus on the T790M mutation, and the proportion of patients with CNS metastases. The data from all recruited patients in the REFLECT study were reported previously [13]. Here we present the data on the Polish population.

### 2.2. Participants

The study comprised adult patients with a confirmed diagnosis of locally advanced unresectable or metastatic NSCLC with confirmed *EGFR* mutation who received a first- or second-generation EGFR-TKI as first-line treatment initiated between 1 January 2015 and 30 June 2018. Exclusion criteria included participation in an interventional clinical trial for an experimental treatment related to NSCLC with *EGFR* mutations at any time and receiving any systemic therapy for locally advanced or metastatic NSCLC before first-line EGFR-TKI therapy. The status of smoking was assessed at the moment of diagnosis of NSCLC based on medical history records. Patients with missing key study data were also excluded.

### 2.3. Data Sources 

We have enrolled consecutive eligible patients and identified them in chronological order starting with the first-line EGFR-TKI therapy within the study entry window. We have analyzed medical records from the initial diagnosis of NSCLC until death or the last available medical record. All patient data were anonymized.

### 2.4. Ethics

This study was performed in compliance with the Declaration of Helsinki, Good Clinical Practice guidelines, and local regulations. The study protocol was approved by the Ethics Committee.

### 2.5. Statistical Analysis

We have used descriptive statistics to describe the results. We have analyzed patients’ demographic and clinical characteristics, molecular testing, and treatment patterns. For categorical variables, frequencies and proportions were used. For continuous variables, the mean, standard deviation, median, and range were used. Point estimations were provided with a 95% confidence interval (CI). Median rwPFS and OS with 95% CI was assessed using the Kaplan–Meier method. Alive patients at the last date of available follow-up were censored. 

## 3. Results

### 3.1. Baseline Characteristics

The study involved 110 patients from six institutions in Poland. All patients met the inclusion criteria, and no patients were excluded from the per-protocol (PP) population. The mean age of patients was 65.99 ± 11.44 years (range 35.0–89.0). Most patients were women (*n* = 70; 63.6%) and 49 (44.5%) patients declared they had never smoked. The most frequent tumor histology was adenocarcinoma, confirmed in 105 (95.5%) patients, and 81 (73.6%) patients had metastatic disease at initial diagnosis. Patients initially at an early or limited regional stage had disease relapse before starting EGFR-TKI treatment. The most common sites of metastases were the lungs (*n* = 45; 40.9%) and bones (*n* = 34; 30.9%). The type of specimen for primary EGFR mutation testing were tissue biopsy (*n* = 80; 72.7%), cytology specimen (*n* = 26; 23.6%), and liquid biopsy (*n* = 4; 3.6%). T90M testing was performed using liquid biopsy (*n* = 44; 75.9%), tissue biopsy (*n* = 9; 15.5%), and cytology specimen (*n* = 9; 8.6%). The type of test conducted to assess T790M mutation status was Cobas EGFR mutation test (*n* = 34; 58.6%), other tests were used in 24 patients (41.4%). Table 1 provides detailed baseline characteristics of the investigated population. 

### 3.2. Characteristics of First-Line Therapy

Afatinib was used as first-line EGFR-TKI therapy in 45 (40.9%) patients, erlotinib in 41 (37.3%) patients, and gefitinib in 24 (21.8%) patients. The first-line EGFR-TKI therapy was discontinued in 90 (81.8%) patients, while it was ongoing at the end of the study for 20 (18.2%) patients. We noticed 85 (77.3%) progression events (radiological progression in 62 patients) (Figure 1). The median rwPFS on first-line EGFR-TKI therapy was 12.9 months (95% CI 10.3–15.4). The PFS rate at 2 years was 17% (95% CI 10–25). Table 2 provides detailed characteristics of first-line EGFR-TKI therapy. PFS on first-line EGFR-TKI therapy is presented in Figure 1. 

### 3.3. Characteristics of Second-Line Therapy

A total of 54 patients started second-line therapy, which was 49.1% of the PP population. All patients starting second-line therapy had progressed on first-line EGFR therapy. Osimertinib was administered to 31 (57.4%) patients, whereas chemotherapy was used in 23 (42.6%) patients. The most common chemotherapy was based on pemetrexed (*n* = 17; 31.5%). The second line of therapy was discontinued in 39 (72.2%) patients, with the most common reason for discontinuation being progression (*n* = 19; 35.2%). The median time to initiation of second-line therapy following discontinuation of first-line therapy was 1.8 months (95% CI 1.0–2.2). Table 3 provides detailed characteristics of second-line therapy.

### 3.4. EGFR Mutation Characteristics

The median time between initial NSCLC diagnosis and *EGFR* mutation testing was 0.62 months Exon 19 deletion was found in 56 (50.9%) patients, the L858R mutation was detected in 42 (38.2%) patients, and other EGFR mutations were found in 12 (10.9%) patients.

Among patients progressing on first-line EGFR-TKI therapy (*n* = 85), only 58 (68.2%) were tested for the T790M mutation. Positive results for the T790M mutation were obtained from 31 (53.4%) of the tested patients (Figure 2). All 31 patients with positive T790M test results received osimertinib in the next lines of therapy. None of the 27 patients with negative T790M test results or the 27 patients without T790M testing received osimertinib in the next lines of therapy. Table 4 provides detailed characteristics of EGFR testing.

### 3.5. Brain Metastasis

Brain metastases were found in 30 (27.3%) patients at any time. Among them, 21 (19.1%) had brain metastases at the start of first-line EGFR-TKI therapy and 9 (8.2%) developed brain metastases during treatment. For those who developed brain metastases during treatment, the median time between the start of first-line EGFR-TKI therapy and the first diagnosis of brain metastases was 22.21 months (6.0–30.7). The median time to death from the first diagnosis of brain metastases (for the total brain metastases population) was 15.5 months (95% CI 9.9–18.0). 

### 3.6. Overall Survival

Among all patients, 60 (54.5%) deaths occurred, and there were 50 (45.5%) censored cases (patients who were alive according to last vital status). The median time to death from the start of first-line EGFR-TKI therapy was 26.2 months (95% CI 18.0–29.7). Estimated probabilities of OS from the start of first-line EGFR-TKI therapy are presented in Table 5 and Figure 3.

## 4. Discussion

The purpose of the REFLECT study was to collect real-world data concerning patients with *EGFR*-mutated advanced NSCLC in Europe and Israel. In the presented manuscript, we described the characteristics of the Polish population of patients with *EGFR*-mutated advanced NSCLC focusing on baseline characteristics of the patients, treatment used in first- and second-line therapy, and *EGFR* mutation testing practice in Poland. 

Lung cancer is the most common malignancy in Poland. In 2012, Poland ranked the seventh in terms of lung cancer mortality among European Union countries [16]. The Polish healthcare system is based on national health insurance, so funding restrictions limits access to services and novel therapies. Reimbursement of new technologies is often delayed relative to Western countries. This could have a direct impact on patient survival in Poland. Osimertinib was introduced for first-line treatment of NSCLC in Poland 2.5 years later than the EMA approval. 

In the presented study, the most common first-line EGFR-TKI was afatinib, which was used in 40.9% of Polish patients. Similar results were observed in the full population of the REFLECT study, in which 45% of patients received afatinib [13]. In other real-world studies, the most common first-line EGFR-TKI was gefitinib [17,18,19] or erlotinib [14,20]. These differences might result from the different locations of the studies and the reimbursement status of the therapies in different countries. 

Due to the inclusion criteria of the study, all patients had a confirmed *EGFR* mutation, the most common of which was exon 19 deletion, similar to other studies [17,19,20]. European Society for Medical Oncology (ESMO) Practice Guidelines and the American Society of Clinical Oncology (ASCO) recommend T790M testing in patients with disease progression after first- or second-generation EGFR-TKIs [21,22]. In the presented study, first-line EGFR-TKI therapy was stopped for 81.8% of patients, with progression being the most common reason (69.1% of patients). In the Polish population, 68.2% of the patients who progressed on first-line EGFR-TKI treatment were tested for T790M, which is similar to the general testing rate in the REFLECT study. This means that 31.8% of patients with disease progression had no opportunity to receive the most effective treatment. A total of 53.4% of tested patients had positive T790M test results, all of whom received osimertinib in the next lines of therapy. In the whole population of the REFLECT study, 71% of patients were tested after progression for T790M, of whom 58% had positive results. Nearly all T790M-positive patients (95%) were treated with osimertinib in the next line of therapy [13]. Based on the results of other studies, T790M testing rates after progression on first-line therapy might affect the number of patients receiving osimertinib during second-line treatment. Shah et al. reported that 74% of patients who progressed on first-line therapy were tested for T790M, 50% of whom had positive results. Of the patients whose tumor tested as T790M positive, 75% subsequently received osimertinib [23]. In the German study performed by Magios et al., T790M testing was performed in 86% of progressive cases, with 53% of cases testing positive. Most patients (94%) with the T790M mutation received osimertinib in the next line of therapy [24]. To the best of our knowledge, our study provides the first description of T790M testing rates in Eastern Europe.

Despite the small number of head-to-head trials with osimertinib and other EGFR-TKIs, we can compare the median PFS of patients receiving first- and second-generation TKIs with osimertinib. In phase III randomized clinical trials, the median PFS for gefitinib and erlotinib was 10–11 months [25,26,27]. In the study by Sequist et al., the median PFS for afatinib was 11.1 months [25]. The results of the meta-analysis performed by Batson et al. revealed the superiority of gefitinib, erlotinib, and afatinib over chemotherapy in the first-line treatment of patients with *EGFR*-mutated NSCLC (26). In the FLAURA study, the median PFS was significantly longer (*p* < 0.001) in patients who received osimertinib (18.9 months) than in patients receiving first-line EGFR-TKI treatment (10.2 months). The median duration of response was 17.2 months with osimertinib and 8.5 months for standard EGFR-TKI treatment [23]. The OS in the osimertinib group was 38.6 months compared to 31.8 months in the standard EGFR-TKI group (*p* = 0.046) [19]. Real-world data on the clinical outcomes of osimertinib in the first-line treatment of NSCLC are limited. The results of the recent study by Shiozawa et al. indicated that osimertinib is effective as a first-line treatment for advanced or recurrent NSCLC in the real-world clinical setting. The estimated median PFS was 17.1 months [27]. Lorenzi et al. reported a median PFS of 18.9 months [28]. Yamamoto et al. reported a median PFS of 19.4 months in patients aged ≥75 years who received osimertinib as first-line treatment [29]. In all of these real-world studies, the median OS was not reached. However, the median PFS was similar to that in the FLAURA study. 

In the current study, the median PFS on first-line EGFR-TKI therapy was 12.9 months, which is longer than in patients receiving first-generation EGFR-TKIs in the FLAURA study, but shorter than in patients who received osimertinib. This may be the result of differences in treatment received, patient characteristics, mutation types, or geographical or ethnical dissimilarity. 

Cumulative OS in our group (26.2 months) was lower than in both studied groups in the FLAURA study. However, patients treated with first- and second-generation EGFR-TKIs, osimertinib, and chemotherapy were included in the OS analysis. As described earlier, a large proportion (31.8%) of patients with disease progression after first-line treatment were not tested for the T790M mutation and did not receive highly effective therapy with osimertinib in the subsequent line of treatment, which could directly affect the OS. Additionally, the median time of 1.8 months between first- and second-line therapy in our study could be described as a delay of highly effective treatment that could limit the overall benefit of the therapy.

In our study, the most common specimen tested for T790M was liquid biopsy, while the most common molecular test was Cobas. In routine clinical diagnostics, molecular analyses on liquid biopsy are mainly based on the detection of mutations of circulating tumor DNA [30]. According to the literature, sensitivity and septicity of the Cobas platform in detecting T790M are 73% and 67% respectively [31].

Brain metastases in patients with advanced NSCLC affect prognosis and have a negative impact on quality of life. Heon et al. showed a reduced risk of CNS progression after treatment with first- and second-generation EGFR-TKIs compared to standard chemotherapy [32]. However, the effectiveness of these drugs is impaired due to the limited ability to penetrate the blood–brain barrier [33,34]. Osimertinib has improved blood–brain barrier penetration and offers clinical benefits in preventing or delaying CNS metastases. Osimertinib has confirmed activity against brain metastases in patients who progressed on first- and second-generation EGFR-TKIs [35,36]. In patients with CNS metastases treated with osimertinib in the FLAURA study, median PFS was not reached compared to 13.9 months in the standard EGFR-TKIs arm (*p* = 0.014). CNS progression was reported in 20% of patients receiving osimertinib compared to 39% of patients receiving standard EGFR-TKIs. Risk analysis showed that the probability of CNS progression was 5% at 6 months in the osimertinib arm versus 18% in the standard EGFR-TKIs arm, and 8% at 12 months in the osimertinib arm versus 24% on standard EGFR-TKIs [37]. In our study, 21 patients had brain metastasis at the start of first-line EGFR-TKI therapy, of whom 76.2% died during the study. Whole-brain radiation therapy was the most common approach in patients with brain metastases, used in 53.3% of patients [38,39]. *EGFR* mutations increase the risk of CNS progression and brain metastasis [40]. Lack of or late introduction of treatment able to cross the blood–brain barrier might lead to loss of effective treatment of metastasis and CNS protection. In the presented study, the median time to death from the first diagnosis of brain metastases was 6.4 months for patients who developed brain metastases during treatment. For patients with brain metastases at baseline, the median time to death was 15.5 months, but this could be the result of treatments including chemotherapy and osimertinib in subsequent lines of therapy. In the FLAURA study, median follow-up for OS of patients with brain metastases in the osimertinib arm was 18.6 months compared to 17.4 months in the standard EGFR-TKI arm [37].

According to the cost-effectiveness analysis by Giuliani and Bonetti, osimertinib is more cost-effective than the other first- and second-generation EGFR-TKIs for first-line treatment of patients with advanced NSCLC with *EGFR* mutations [41].

The main limitation of this study is its observational and retrospective character. The study was descriptive with no formal hypothesis on treatment efficacy. 

## 5. Conclusions

The results of this descriptive analysis of the Polish population of the REFLECT study highlight the need for effective treatment of patients with advanced, *EGFR*-mutated NSCLC. First-line therapy was based on first- and second-generation EGFR-TKIs. The median PFS of patients on first-line therapy was lower than that of patients receiving osimertinib during the FLAURA study and in available real-world evidence studies. Brain metastases were a negative prognostic factor. First- and second-generation EGFR-TKIs were not sufficiently effective in the treatment and prevention of CNS progression, likely due to poor penetration of the blood–brain barrier. Considering that nearly one-third of patients with disease progression after first-line EGFR-TKI therapy were not tested for the T790M mutation and did not have the opportunity to receive effective treatment, implementation of osimertinib as first-line therapy is the preferred option [42]. 

## Figures and Tables

**Figure 1 cancers-15-01581-f001:**
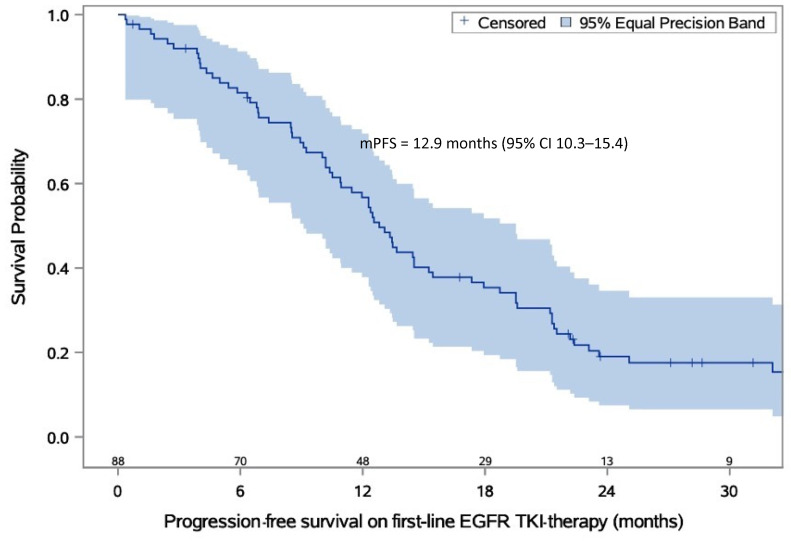
Estimated probability of progression-free survival on first-line EGFR-TKI therapy. The numbers above the *x*-axis describe the number of progression-free patients. Abbreviations: EGFR-TKI—epidermal growth factor receptor tyrosine kinase inhibitor; mPFS—median progression-free survival.

**Figure 2 cancers-15-01581-f002:**
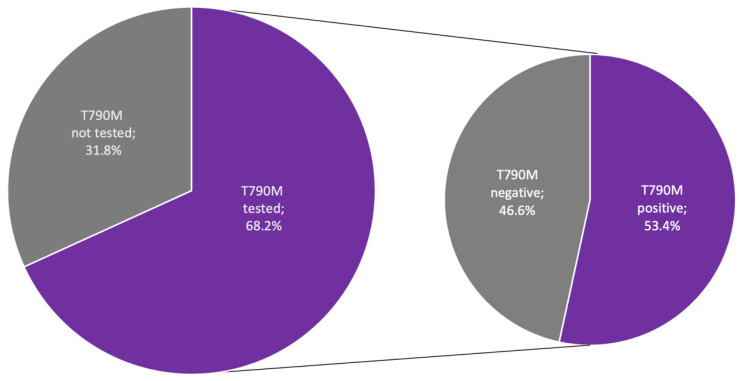
T790M testing patterns in patients with progression on first-line EGFR-TKI therapy. Abbreviation: EGFR-TKI—epidermal growth factor receptor tyrosine kinase inhibitor.

**Figure 3 cancers-15-01581-f003:**
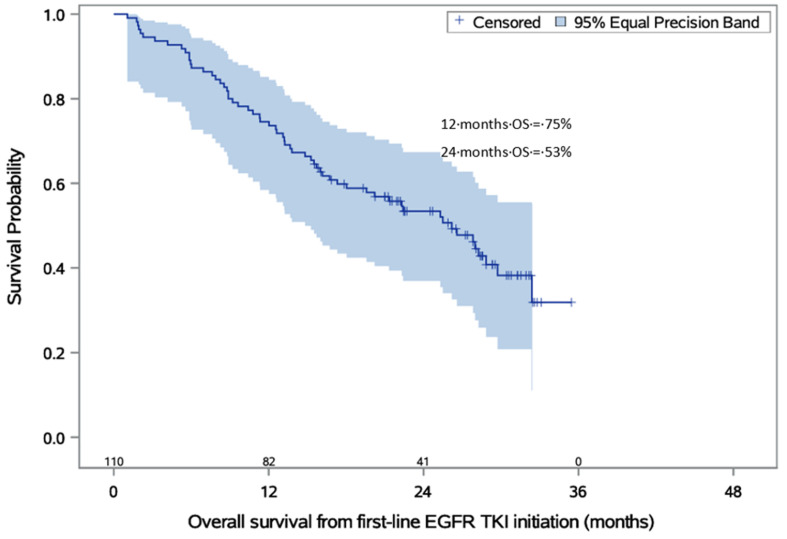
Estimated probabilities of overall survival from the start of first-line EGFR-TKI therapy. The numbers above the *x*-axis describe the number of alive patients. Abbreviations: EGFR-TKI—epidermal growth factor receptor tyrosine kinase inhibitor; OS—overall survival; mOS—median overall survival.

**Table 1 cancers-15-01581-t001:** Baseline characteristics of patients (*n* = 110).

Age	Years
Mean (SD)	65.9 (11.44)
Gender	*n* (%)
Female	70 (63.6)
Male	40 (36.4)
Smoking status at initial diagnosis	*n* (%)
Current smoker	6 (5.5)
Former smoker	29 (26.4)
Never smoker	49 (44.5)
Not known	26 (23.6)
Histology at initial diagnosis	*n* (%)
Adenocarcinoma	105 (95.5)
Mixed histology	3 (2.7)
Other	2 (1.8)
Tumor stage at initial diagnosis (Stage)	*n* (%)
Early stage (I)	12 (10.9)
Limited regional (II)	5 (4.5)
Locally advanced (IIIA/B)	12 (10.9)
Metastatic (IV)	81 (73.6)
Site of distant metastases	*n* (%)
Adrenal	9 (8.2)
Bone	34 (30.9)
Brain	20 (18.2)
Liver	13 (11.8)
Lung	45 (40.9)
Pleura	32 (29.1)
Other	13 (11.8)

Abbreviations: SD—standard deviation.

**Table 2 cancers-15-01581-t002:** Characteristics of first-line EGFR-TKI therapy.

Type of First-Line EGFR-TKI Therapy	*n* (%)
Afatinib	45 (40.9)
Erlotinib	41 (37.3)
Gefitinib	24 (21.8)
Discontinuation of first-line EGFR-TKI therapy	*n* (%)
Yes	90 (81.8)
No	20 (18.2)
Reason for discontinuation of first-line EGFR-TKI therapy	*n* (%)
Radiological progression	62 (56.4)
Clinical progression	14 (12.7)
Death	8 (7.3)
Other reason	6 (5.5)
PFS on first-line EGFR-TKI therapy	Months
Minimum time to event	1.0
Maximum time to event	23.0
Median (95% CI)	12.9 (10.3–15.4)

Abbreviations: CI—confidence interval; EGFR-TKI—epidermal growth factor receptor tyrosine kinase inhibitor; PFS—progression-free survival.

**Table 3 cancers-15-01581-t003:** Characteristics of second-line therapy.

Type of Second-Line EGFR-TKI Therapy	*n* (%)
Osimertinib	31 (57.4)
Chemotherapy	23 (42.6)
Cytotoxic agents	*n* (%)
Cisplatin/carboplatin + pemetrexed	15 (27.8)
Carboplatin + paclitaxel	1 (1.9)
Vinorelbine	4 (7.4)
Gemcitabine	1 (1.9)
Discontinuation of second-line therapy	*n* (%)
Yes	39 (72.2)
No	15 (27.8)
Reason for discontinuation of second-line therapy	*n* (%)
Radiological progression	19 (35.2)
Clinical progression	5 (9.3)
Death	5 (9.3)
Other reason	9 (16.7)
Not known	1 (1.9)
Time between the start of first-line EGFR-TKI therapy and the start of second-line therapy	Months
Mean (SD)	13.87 (6.47)
Median	13.57
Minimum, Maximum	1.3, 26.7

Abbreviations: EGFR-TKI—epidermal growth factor receptor tyrosine kinase inhibitor; SD—standard deviation.

**Table 4 cancers-15-01581-t004:** Characteristics of *EGFR* mutations.

Time between Initial NSCLC Diagnosis and *EGFR* Mutation Testing	Months
Mean (SD)	4.28 (11.75)
Median	0.62
Type of identified *EGFR* mutation	*n* (%)
Exon 19 deletion	56 (50.9)
L858R mutation	42 (38.2)
Other *EGFR* mutation	12 (10.9)

Abbreviations: EGFR—epidermal growth factor receptor; SD—standard deviation.

**Table 5 cancers-15-01581-t005:** Overall survival.

Time to Death from First Diagnosis of Locally Advanced or Metastatic Disease	Months
Minimum time to event	2.0
Maximum time to event	33.0
Median (95% CI)	27.3 (18.1–31.9)
Time to death from start of first-line EGFR-TKI therapy	Months
Minimum time to event	1.0
Maximum time to event	32.0
Median (95% CI)	26.2 (18.0–29.7)

Abbreviations: CI—confidence interval; EGFR-TKI—Epidermal growth factor receptor tyrosine kinase inhibitor.

## Data Availability

The datasets generated during and/or analyzed during the current study are available from the corresponding author on reasonable request.

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
