# Peer review of "Treatment Patterns, Testing Practices, and Outcomes in Patients with EGFR Mutation-Positive Advanced Non-Small-Cell Lung Cancer in Poland: A Descriptive Analysis of National, Multicenter, Real-World Data from the REFLECT Study"

_cancers, 2023, doi:10.3390/cancers15051581_

Round 1

Reviewer 1 Report

this study provides real world data for the management of EGFR mutant patients from Poland prior to the introduction of first line osimertinib; this may also hint at what Eastern European practice may be.

The methodology and statistical analyses are appropriate, and the results confer what is expected from 1st and 2nd gen EGFR TKIs re PFS, and reveal that care is not optimal in terms of testing at progression for T790M.

Quoting PFS according to 1st vs 2nd gen EGFR TKI (afatinib) would have also been interesting.

Author Response

Thank you for your feedback. We appreciate the insights you have provided and have taken note of your suggestions. We agree that comparing PFS according to first versus second generation EGFR TKIs would be interesting and are exploring ways to incorporate this into our analysis. Unfortunately, we only have the collective PFS results for various EGFR-TKIs.

Reviewer 2 Report

In read with interest the research article entitled “Treatment patterns, testing practices, and outcomes in patients with EGFR mutation-positive advanced non-small-cell lung cancer in Poland: a descriptive analysis of national, multicenter, real-world data from the REFLECT study”. Adam Pluzanski and colleagues report a real-world experience on the Polish patients subgroup of an international multicentre onservational study, namely REFLECT study, performed on EGFR-mutated NSCLC patients.

This is a very interesting study, reporting indications on drugs used as first- and second-line therapies for EGFR-mutated NSCLC in a national subgroup of patients. This reviewer only has a few suggestions:

- In the Discussion paragraph, the authors state that “we describe……EGFR mutation testing practice in Poland”. I would suggest to add details about the EGFR testing: which molecular testing techniques were used, and using which type of biological sample (FFPE biopsies, cytological smears, liquid biopsy) both for activating mutations and T790M mutations. In this context, please add a comment on sensitivity and specificity of techniques used (PMID 36356494, 31857950). Moreover, since lung cancer is often characterized by genetic heterogeneity, also regarding the EGFR status, please add a comment whether more than one bioptic sample was used, both for activating and resistance mutations (PMID 27133750, 33067323).

- In the introduction, the authors state that “For patients with locally advanced and/or metastatic NSCLC harboring an EGFR mutation, the first-line treatment includes first-generation EGFR-tyrosine kinase inhibitors (TKIs) such as er-lotinib and gefitinib, second-generation EGFR-TKIs such as afatinib and dacomitinib, and the newest third-generation EGFR-TKI osimertinib”.

Osimertinib is now the standard of care of EGFR-mutated NSCLC, please include a comment about this change of clinical indications, referring to the use of first- and second-generation TKIs as the standard of care during REFLECT study enrolment (2015, 1st January- 2018, 30th June) [PMID 25923549, 33186860].

In the Discussion, the authors state that “In the current study, the median PFS on first-line EGFR-TKI therapy was 12.9 months, which is longer than in patients receiving first-generation EGFR-TKIs in the FLAURA study, but shorter than in patients who received osimertinib. This may be result of differences in patient characteristics, mutation types or geographical or ethnical dissimilarity”.

This is true in part. The main reason for this difference is manly dependent on the different treatment administered as a first-line therapy, as FLAURA trial enrolled previously untreated EGFR-mutated patients, demonstrating that osimertinib strongly prolonged PFS; the authors should comment this as a normal consequence of different treatments, not as a study limitation [PMID 29151359].

Minor concerns:

- In Table 1, please indicate median with max and min, or mean with standard deviation; there in no adding information on reporting both information; Please also change “sex” in “gender”; please add a definition for former smokers in materials and methods.

- In Figure 1, please add the Confidence interval in the graph or in the legend.

Author Response

-Thank you for your time and valuable comments on our manuscript. We hope that we have improved our manuscript enough for publication. In the attached revised manuscript, we have included more specific information. We responded to all questions and suggestions. All changes are marked in red.

In read with interest the research article entitled “Treatment patterns, testing practices, and outcomes in patients with EGFR mutation-positive advanced non-small-cell lung cancer in Poland: a descriptive analysis of national, multicenter, real-world data from the REFLECT study”. Adam Pluzanski and colleagues report a real-world experience on the Polish patients subgroup of an international multicentre onservational study, namely REFLECT study, performed on EGFR-mutated NSCLC patients.

This is a very interesting study, reporting indications on drugs used as first- and second-line therapies for EGFR-mutated NSCLC in a national subgroup of patients. This reviewer only has a few suggestions:

In the Discussion paragraph, the authors state that “we describe……EGFR mutation testing practice in Poland”. I would suggest to add details about the EGFR testing: which molecular testing techniques were used, and using which type of biological sample (FFPE biopsies, cytological smears, liquid biopsy) both for activating mutations and T790M mutations. In this context, please add a comment on sensitivity and specificity of techniques used (PMID 36356494, 31857950). Moreover, since lung cancer is often characterized by genetic heterogeneity, also regarding the EGFR status, please add a comment whether more than one bioptic sample was used, both for activating and resistance mutations (PMID 27133750, 33067323).

-Thank you for this advice. We have added details about tested samples and testing techniques to the results section. Also, we have added relevant paragraph to the discussion and provided new references.

 In the introduction, the authors state that “For patients with locally advanced and/or metastatic NSCLC harboring an EGFR mutation, the first-line treatment includes first-generation EGFR-tyrosine kinase inhibitors (TKIs) such as er-lotinib and gefitinib, second-generation EGFR-TKIs such as afatinib and dacomitinib, and the newest third-generation EGFR-TKI osimertinib”.

Osimertinib is now the standard of care of EGFR-mutated NSCLC, please include a comment about this change of clinical indications, referring to the use of first- and second-generation TKIs as the standard of care during REFLECT study enrolment (2015, 1st January- 2018, 30th June) [PMID 25923549, 33186860].

-Thank you for this suggestion, we have changed this paragraph.

In the Discussion, the authors state that “In the current study, the median PFS on first-line EGFR-TKI therapy was 12.9 months, which is longer than in patients receiving first-generation EGFR-TKIs in the FLAURA study, but shorter than in patients who received osimertinib. This may be result of differences in patient characteristics, mutation types or geographical or ethnical dissimilarity”.

This is true in part. The main reason for this difference is manly dependent on the different treatment administered as a first-line therapy, as FLAURA trial enrolled previously untreated EGFR-mutated patients, demonstrating that osimertinib strongly prolonged PFS; the authors should comment this as a normal consequence of different treatments, not as a study limitation [PMID 29151359].

-Thank you for this suggestion, we have modified this paragraph.

Minor concerns:

- In Table 1, please indicate median with max and min, or mean with standard deviation; there in no adding information on reporting both information; Please also change “sex” in “gender”; please add a definition for former smokers in materials and methods.

-Thank you for this advice, we have corrected it as suggested. The status of smoking was assessed at the moment of diagnosis of NSCLC based on medical history records (we have added it to the material and methods section). No time period of abstinence was specified to select former smokers. 

In Figure 1, please add the Confidence interval in the graph or in the legend.

-We have added 95% CI to the graph.